# Oligomicrodontia in a Pediatric Cancer Survivor after Chemotherapy: A Case Report

**DOI:** 10.3390/healthcare10081521

**Published:** 2022-08-12

**Authors:** Ana Zulijani, Martina Žigante, Luka Morelato, Berislav Perić, Ana Milardović

**Affiliations:** 1Department of Oral Surgery, Clinical Hospital Center Rijeka, 51000 Rijeka, Croatia; 2Department of Orthodontics, Clinical Hospital Center Rijeka, 51000 Rijeka, Croatia; 3Department of Oral Surgery, Faculty of Dental Medicine, University of Rijeka, 51000 Rijeka, Croatia; 4Department of Oral and Maxillofacial Surgery, Dubrava University Hospital, 10000 Zagreb, Croatia; 5School of Dental Medicine, University of Zagreb, 10000 Zagreb, Croatia; 6Pediatric Intensive Care Unit, Clinical Hospital Center Rijeka, 51000 Rijeka, Croatia; 7Faculty of Medicine, University of Rijeka, 51000 Rijeka, Croatia

**Keywords:** pediatric cancer survivors, late effects, microdontia, dental development disorders, dental care, case report

## Abstract

Chemotherapy used on pediatric patients especially those below 3 years of age causes disturbances in dental development. The aim of this case report was to present the late dental effects of chemotherapy in a patient treated for anaplastic ependymoma (WHO III) at an early age. Radiographic findings at the age of 9 years showed oligomicrodontia of six teeth, maxillary lateral incisors, and maxillary and mandibular first premolars. Pediatric cancer survivors after chemotherapy have an increased risk of one or more dental development disorders. To ensure proper dental care and to assess the long-term effects on oral health, tooth development, and occlusion, the involvement of a dentist is crucial. Adequate diagnosis and well-planned treatment of the dental defect can significantly improve patient oral health-related quality of life.

## 1. Introduction

The greater improvement in diagnostics and application of multimodal therapies for childhood malignant diseases has resulted in a high 5-year survival rate, more than 80% depending on the type of malignancy [1]. Despite attempts to find anticancer drugs that only affect tumor cells, chemotherapy is still the treatment of choice with all toxic effects on normal, actively proliferating cells [2]. As is well known, cancer chemotherapy treatment in children can cause a wide range of long-term side effects on cardiovascular, pulmonary, endocrine, and other systems.

Oeffinger et al. found that antineoplastic treatment in children results in at least one late effect on any organ, impacting around 60–90% of patients in adult age [3]. Because of this, young children are usually followed up by a multidisciplinary team. Unfortunately, dental and maxillofacial professionals are not often included, even though dental and maxilla–mandibular anomalies are described as common long-term side effects of childhood cancer treatment [4,5,6,7].

However, nowadays, a great interest is devoted to esthetic dental appearance, which, if compromised, negatively affects the psychosocial wellbeing and thus the quality of life, indicating a need for adequate attention [8,9].

Effects of chemotherapy on dental development may vary, depending on the time concurrent developmental phase which was affected. Dental development anomalies include arrested root development, taurodontism, microdontia, hypodontia, and enamel defects, while tooth eruption seems to be delayed [4,5,6,10].

Arrested root development is attributed to the use of cytotoxic drugs that decelerate or prevent the formation of Hertwig’s epithelial root sheath, thereby changing the crown-to-root ratio and osteodentin formation which leads to an early apex closure. Similarly, the disturbance of ameloblasts during the period of tooth development by cytotoxicity alters their mitotic activity changing secretory function, membrane permeability, and calcium homeostasis [11].

The subjective sensation of dry mouth is a potential side effect following head and neck radiation but the association of chemotherapy alone with xerostomia is controversial [12]. It can be concluded that the DMFT score (the sum of the number of Decayed, Missing due to caries, and Filled Teeth in the permanent teeth) of chemotherapy patients depends primarily on family history, hygiene patterns, and socioeconomic status [13].

The aim of this report was to present the dental development disturbances following chemotherapy at an early age and to point out the need for routine dental care and the involvement of a dentist in monitoring the long-term effects on oral health, tooth development, and occlusion.

## 2. Case Report

A 9-year-old girl was referred to the Department of Oral Surgery, Dental Clinic of Clinical Hospital Center Rijeka because of disturbances in the eruption of the permanent teeth. Consistent with the ethical requirements of the Faculty of Dental Medicine, University of Rijeka, Croatia, and the Clinical Hospital Center Rijeka, Croatia, written informed consent for the publication of this study was obtained from the patient’s parents.

### 2.1. Anamnestic History

The medical history revealed that at the age of 4 months she was diagnosed with anaplastic ependymoma WHO grade III. Two weeks after total surgical removal of the tumor, chemotherapy protocol HIT2000 was initiated, consisting of four courses of five different drugs (cyclophosphamide, vincristine, methotrexate, etoposide, and carboplatin) administered in five cycles (Figure 1). The interval between the first day of each cycle was 21 days, and between each course 14 days. Drug doses are calculated taking into account the patient’s current body surface area. Considering the young age of the child, 2/3 of the full dose was applied up to 6 months, from 7–12 months of age 4/5 of the full dose, and after one year of age the full dose. More than 0.5 × 10^9^ of granulocytes/L and more than 100 × 10^9^ of platelets/L were mandatory homological criteria for the commencement of a new course. In the case of hematological toxicity, the patient received granulocyte colony-stimulating factor and blood transfusion (21 times). No other unacceptable toxicity occurred during chemotherapy. In the period of maintenance chemotherapy, the patient received three cycle courses of cisplatin, lomustine, and vincristine (Figure 1). After that, due to difficulties with the central venous catheter, the patient underwent treatment with temozolomide, administered in five cycles (first cycle 150 mg/m^2^, second cycle 200 mg/m^2^, third/fourth/fifth cycle 250 mg/m^2^), every four weeks 5 days. Due to the young age, radiotherapy was not applied.

At the time of the first visit to the Department of Oral Surgery, she showed no signs of recurrence on annual MRI follow-up. Currently, she is under the supervision of a pediatric endocrinologist due to premature puberty. The data of physical examination showed normal stature and weight among her peers, that is, around the 75th percentile (Figure 2).

### 2.2. Physical Examination

Extraoral facial analysis shows a symmetrical face, obtuse nasolabial angle, acute mentolabial angle, and straight facial profile with a prominent chin (Figure 3).

### 2.3. Intraoral Examination

Dental history indicates acute oral complications of chemotherapy, mucositis, and previous visits to the dentist for restorations on teeth 54, 64, 75, and extraction of deciduous teeth 52, 62.

Intra-oral examination showed a late mixed dentition stage, fair oral hygiene, and healthy gingiva. The examination revealed the following teeth: 16, 55, 54, 11, 21, 63, 64, 65, 26, 36, 75, 74, 33, 32, 31, 41, 42, 43, 84, 85, and 46 (Figure 3.). All erupted teeth had a regular morphology and structure. Teeth 14, 13, and 12 were initiated to erupt. Her permanent right maxillary lateral incisor appeared abnormal in shape. Caries were present on 75 and restoration on 54 and 64. Orthodontic diagnoses were anterior and unilateral posterior crossbite (Figure 4).

### 2.4. Radiological Examination

The orthopantomogram showed the presence of all permanent teeth at varying stages of development and eruption, but the microdontia of six teeth (Figure 5). In particular, the crowns of all first premolars (both maxillary and mandibular) and maxillary permanent lateral incisors were reduced in size. In the family history, microdontia was not present.

The root dimensions appeared to be normally formed (Figure 5).

Cephalometric analysis revealed a tendency towards maxillary retrognathia, skeletal class III, a counterclockwise rotation of the maxilla, and proclination of lower incisors (Figure 6).

Six microdontic teeth give a Dental Defect Index (DeI) of 24, which at the current patient age, despite DeI, reaches its maximum value after all roots have completed their development at adolescent age [14].

## 3. Discussion

Development disturbances can be influenced by genetic, epigenetics, and environmental factors, such as chemotherapy, radiotherapy, and chemoradiotherapy treatment at an early age. Various studies reported effects on craniofacial and dental development caused by chemotherapy administered during the stages of tooth development [4,6,10,13,14,15].

Dental development is a complex process that begins already in utero and ends approximately around 15 years with the eruption of permanent dentition. First molars, maxillary central incisors, mandibular incisors, and canines are the first permanent teeth that begin the development process in the period from the time of birth to 4–5 months. The maxillary lateral incisors are the next teeth that start the development process at the age of 10–12 months, followed by the first premolars from 18–24 months. The range of mineralization of second premolars and molars is between 24 and 36 months. The third molars are the last teeth that develop, between the ages of 7–10 years [16].

In this case, the presented oligomicrodontia of six teeth is probably caused by chemotherapy for anaplastic ependymoma which was received at the period of maxillary lateral incisors and first premolar mineralization, between five to twenty-two months of age. This assumption about the importance of age during chemotherapy is in line with previous research, which showed the higher prevalence of microdontia in patients younger than 3 years (75%), treated with stem cell transplantation after high-dose chemotherapy and/or total body irradiation [17].

The incidence of microdontia in healthy populations is very low, rarely estimated to range from 0.8% to 2.5% in orthodontic populations, often limited to maxillary lateral incisors, “peg-shaped” teeth [17,18,19]. Microdontia more often occurs in relation with tooth agenesis (hypodontia), in 17.7% of the patients [20].

Truly generalized microdontia is usually associated with syndromic conditions, Gorlin–Chaudhry–Moss syndrome, Williams’s syndrome, orofaciodigital syndrome (type 3), oculo-mandibulo-facial syndrome, pituitary dwarfism, and others [21].

Children treated for cancer at an early age have been found to have a prevalence of microdontia from 13.5% up to 78%, depending on treatment modality [16,17,18].

To the best of our knowledge, this is the second case with the reduced size (microdontia) of six permanent teeth after chemotherapy received at an early age reported in the literature [22]. Knowing the terms used for hypodontia of 1–5 missing teeth, and oligodontia used for six or more missing teeth, we suggest using the term oligomicrodontia for six or more microdontic teeth in nonsyndromic patients.

Smaller teeth especially in the frontal part of the jaws can be aesthetically problematic which can negatively affect oral health-related quality of life (OHRQoL). Additionally, microdontia can have an effect on the difficulty to achieve normal occlusion and is associated with mastication [23,24,25]. Besides the aesthetic concern, microdontic teeth complicate the orthodontic treatment plan since some orthodontic appliances cannot be bonded on such teeth because of their fragility. Additionally, the amount of the orthodontic force should be considered if applied on such teeth.

Malocclusion as a result of altered craniofacial development can also be caused by hypodontia. In this case, hypodontia was not observed which can suggest that genetic background plays an important role, although some studies have described tooth agenesis in cancer survivors treated with chemotherapy. Hypodontia, development agenesis of one or more teeth, is the most common dental development anomaly with a prevalence of 2 to 10% in a healthy population, usually associated with genetic background [17,26]. Proc et al. observed that in the cancer survivors, hypodontia was three times more frequent, around 31.14% [18]. Hypodontia in cancer survivors is mostly correlated with radiotherapy during dental development [17].

In this case, the adverse effect of chemotherapy on root development is only visible in microdont teeth, probably as a consequence of the reduced dimensions of the crown. In the study on 97 acute lymphoblastic leukemia survivors treated only with chemotherapy, Sonis et al. reported arrested root development in 63% of patients [27]. A retrospective evaluation of orthopantomograms of 70 childhood cancer survivors with a mean age of 4.17 years at diagnosis reported 62% of root defects, most commonly impaired root growth, and treated with radiotherapy [28]. Observing different root development disorders, the prevalence reaches an even higher rate, 77% [29]. Due to the young patient’s age presented in this case, not all roots were developed so we cannot discuss and conclude that no arrested root development will happen in the future. It can be stated that young chemotherapeutic patients must be followed by dental professionals intensively to the end of tooth eruption and root development.

More than 300 genes are known to be engaged in tooth development. Mutations in the sequence of a gene or group of genes are related to dental development disorders. In addition, epigenetic and environmental factors can cause changes in gene expression or interfere with their protein function. Arrested tooth development can cause several environmental factors such as chemotherapy agents and irradiation [30].

Although dental development disturbances have been associated with exposure of various chemotherapeutic agents, the molecular mechanisms are poorly understood. Chemotherapy treatment can affect dental development by direct toxicity to odontogenic cells or by disrupting signaling pathways between ectodermal and/or mesenchymal cells [31,32,33,34].

Experimental studies on animals have shown the interference of alkylating agents with the process of dentinogenesis by binding to DNA in the S-phase of mitosis, which ultimately resulted in the early apoptosis of preodontoblasts [6]. In the study, Stolze et al. found a significantly increased risk of one or more dental development disorders in pediatric cancer survivors younger than 3 years old treated with chemotherapy compared with older children as well with the dose exposure of alkylating agent. The dose of >4 g/m^2^ significantly increased the risk of developing ≥ 1 dental abnormalities [10]. An eight times greater risk of developing dental abnormalities was reported by Seremedi et al. in patients who received high doses of cyclophosphamide, >4 g/m^2^ [28]. Likewise, it showed a significant association of topoisomerase 2 inhibitors (etoposide) with one dental development disorder using univariable analysis, while no association with risk was obtained using Poisson analysis [10]. Another commonly used chemotherapeutic agent, vincristine, has been identified as a cause of tooth abnormalities by interfering with the calcium transport mechanism and secretory function of ameloblasts [33]. Other studies have also shown the presence of dental defects with the use of DNA crosslinking agents such as cisplatin and carboplatin [34,35].

The majority of therapeutic protocols, similar to those in this presented case, involve the use of multiple agents, which makes it difficult to attribute the specific effect of a single agent to the described dental defects.

Dental developmental anomalies may affect dental occlusion, but there is a lack of data on pediatric cancer survivors. Although malocclusion was equally common, Proc et al. showed a higher probability of crossbite in patients treated for cancer compared to their healthy peers, as well as a more advanced dental age compared to chronological age [36]. Orthodontic treatment should be considered and planned carefully in patients treated for cancer. The American Academy of Pediatric Dentistry suggested modifications to orthodontic treatment in such patients, such as using lighter forces due to the risk of root resorption, earlier termination of treatment, using simpler methods, and avoiding the treatment of the lower jaw [37].

## 4. Conclusions

Disrupted tooth development can be expected in children treated for cancer, which suggests that dentists should keep in mind that such patients should be regularly monitored even after the end of therapy to minimize and possibly intercept the consequences.

## Figures and Tables

**Figure 1 healthcare-10-01521-f001:**
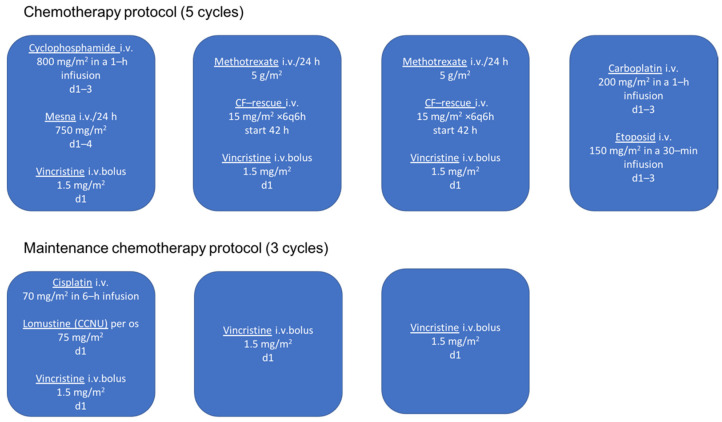
Chemotherapy protocol in patient treated for anaplastic ependymoma (HIT 2000).

**Figure 2 healthcare-10-01521-f002:**
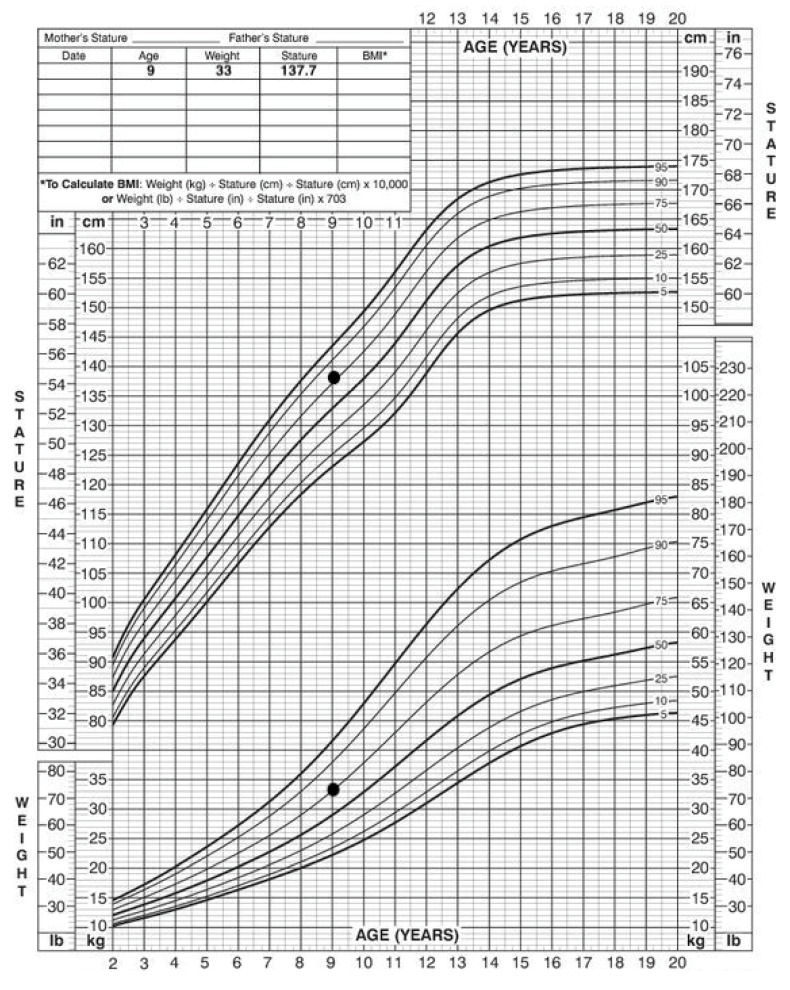
Patient stature and weight, 2 to 20 years: Girls stature-for-age and weight-for-age percentiles. Black marks indicate data of physical examination.

**Figure 3 healthcare-10-01521-f003:**
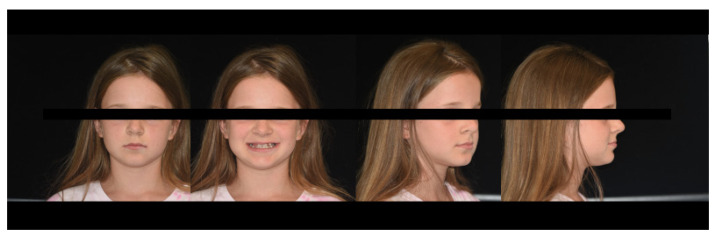
Extraoral facial analysis.

**Figure 4 healthcare-10-01521-f004:**
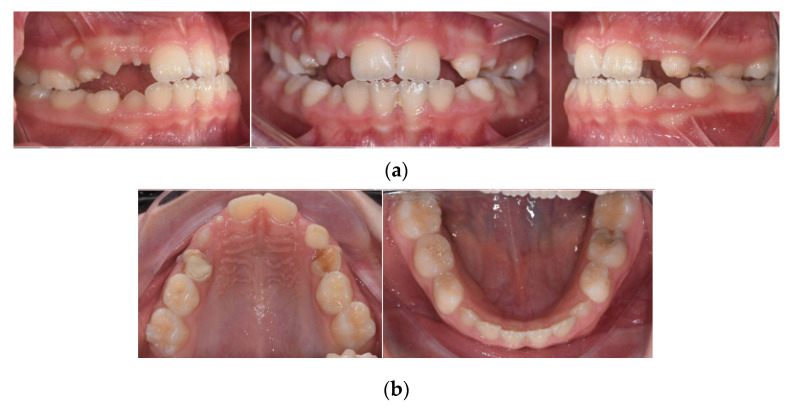
Intraoral clinical photographs, including views of: (**a**) anterior and posterior crossbite; (**b**) upper and lower arch.

**Figure 5 healthcare-10-01521-f005:**
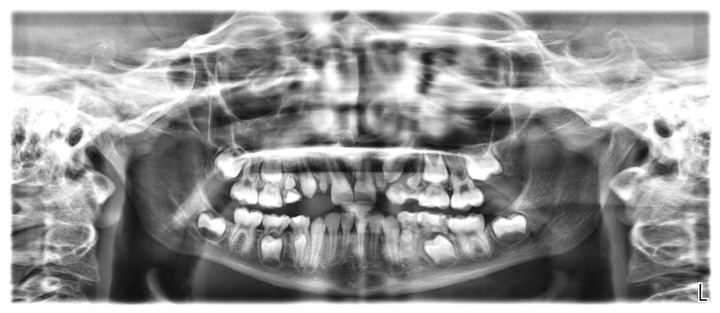
Orthopantomogram of a 9-year-old girl showing reduced size of teeth 14, 12, 22, 24, 34, and 44.

**Figure 6 healthcare-10-01521-f006:**
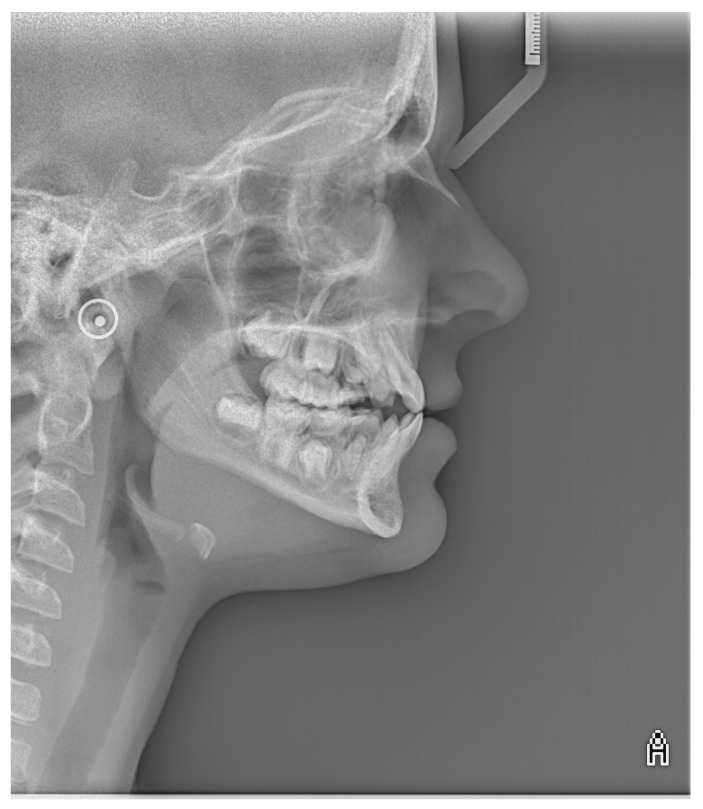
Image of lateral cephalometric radiograph.

## Data Availability

Not applicable.

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
