# Peer review of "Oligomicrodontia in a Pediatric Cancer Survivor after Chemotherapy: A Case Report"

_healthcare, 2022, doi:10.3390/healthcare10081521_

Round 1

Reviewer 1 Report

Many thanks for the submission to this Journal: this paper is interesting with some additions and modifications. In the introduction at line 38 please modify the phrase with:
"Unfortunately, dental and maxillofacial professionals are often not included, even though dental and maxillo-mandibular anomalies are described
as a common long-term side effect of childhood cancer treatment."

and add this citation in the reference:

Giovannetti F, Aboh IV, Chisci G, Gennaro P, Gabriele G, Cascino F, Di Curzio P, Iannetti G. Langerhans cell histiocytosis: treatment strategies. J Craniofac Surg. 2014 May;25(3):1134-6.

Reviewer 2 Report

The manuscript "Oligomicrodontia in pediatric cancer survivor after Chemo-2 therapy: A Case Report" presents the clinical case of a pediatric patient previously diagnosed with WHO grade III anaplastic ependymoma who developed microdontia after treatment with the chemotherapy protocol HIT2000. My comments are the following:

- Given that the administered doses of chemotherapeutic drugs could be involved in the development of microdontia, it is important to mention what these were in the patient described, as well as in the cases described in the discussion.

- It is necessary to describe how it was monitored that the levels of chemotherapeutic drugs were at adequate levels in the patient and that these never exceeded the established levels.

- It would be useful if they hypothesized what is the reason for the development of microdontia (affectation of the stem cells of the teeth, changes in the stroma, DNA damage or increased production of ROS).

- Based on the literature, it should also be hypothesized if it is a single case or it can be developed in what percentage of pediatric cancer survivors.

- Has any case been described in adults?

- There are some minor grammatical errors in the text

Round 2

Reviewer 2 Report

No additional comments